# Wear Characteristics of Superalloy and Hardface Coatings in Gas Turbine Applications–A Review

**Ahmad Afiq Pauzi** [1,*]**, Mariyam Jameelah Ghazali** [2]**, Wan Fathul Hakim W. Zamri** [2] **and Armin Rajabi** [2]

1   Materials Engineering and Testing Group, TNB Research Sdn Bhd, Kajang 43000, Malaysia
2   Centre for Materials Engineering and Smart Manufacturing, Faculty of Engineering and Built Environment, Universiti Kebangsaan Malaysia, UKM Bangi, Selangor 436000, Malaysia; mariyam@ukm.edu.my (M.J.G.); wfathul.hakim@ukm.edu.my (W.F.H.W.Z.); arminrajabi@gmail.com (A.R.)
*   Correspondence: afiqpauzi@tnb.com.my

**Abstract:** In the gas-turbine research field, superalloys are some of the most widely used materials as they offer excellent strength, particularly at extreme temperatures. Vital components such as combustion liners, transition pieces, blades, and vanes, which are often severely affected by wear, have been identified. These critical components are exposed to very high temperatures (ranging from 570 to 1300 °C) in hot-gas-path systems and are generally subjected to heavy repair processes for maintenance works. Major degradation such as abrasive wear and fretting fatigue wear are predominant mechanisms in combustion liners and transition pieces during start–stop or peaking operation, resulting in high cost if inadequately protected. Another type of wear-like erosion is also prominent in turbine blades and vanes. Nimonic 263, Hastelloy X, and GTD 111 are examples of superalloys used in the gas-turbine industry. This review covers the development of hardface coatings used to protect the surfaces of components from wear and erosion. The application of hardface coatings helps reduce friction and wear, which can increase the lifespan of materials. Moreover, chromium carbide and Stellite 6 hardface coatings are widely used for hot-section components in gas turbines because they offer excellent resistance against wear and erosion. The effectiveness of these coatings to mitigate wear and increase the performance is further investigated. We also discuss in detail the current developments in combining these coating with other hard particles to improve wear resistance. The principles of this coating development can be extended to other high-temperature applications in the power-generation industry.

**Keywords:** wear; gas turbine; superalloy; hardface coating

## 1. Introduction

A power-generating gas turbine is designed to run continuously under base-load operation with a major yearly inspection. The hot-gas-path components of a gas turbine consist of high-value and finite-life components [1]. The durability of hot-gas-path components has always been of interest to gas-turbine operators because half of the maintenance and repair costs of gas-turbine units can be attributed to the overhaul manpower and the refurbishment and replacement of these high-value parts [2].

Bohidar et al. [3] found that hot-gas-path components are subjected to extremely high temperatures ranging from 570 to 1300 °C and thus experience simultaneous thermal damages, such as creep, fatigue, and high temperature wear. The materials used in hot-gas-path components are made from various strengthened superalloys with excellent weldability and manufacturability, such as Nimonic 263 and Hastelloy X [4].

A gas turbine has a typical configuration comprising a compressor and turbines fixed together on a single shaft, which is connected to a generator [5]. Compressed air, at a typical pressure of 14 atm, is directed into the combustion section where the fuel is injected and burned and thus reacts with the compressed air. The hot gasses, as the output of this reaction, expand through the turbine section before being exhausted to the atmosphere [6]. Figure 1 shows the main sections of a gas turbine in a typical configuration.

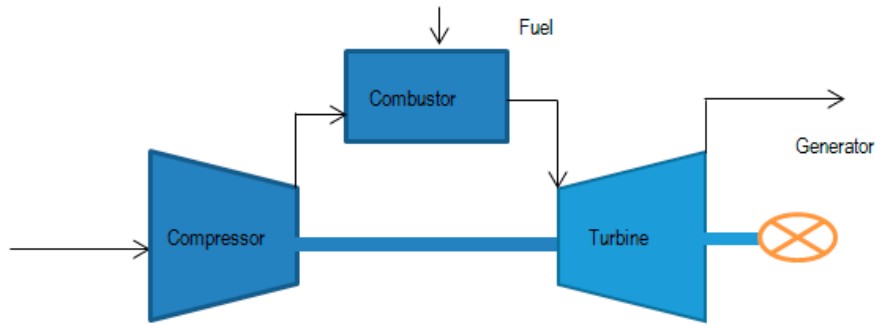

**Figure 1.** Main sections of a gas turbine.

For the operation of power generation gas turbines, the compressor draws compressed air into the combustion section at speeds of 160 km/h. In the combustion system, the fuel is injected into the combustion chamber where it is mixed with the compressed air. The mixture of compressed air and fuel is burned at a temperature of approximately 1300 °C. The combustion that has been produced by the reaction between compressed air and fuel produces high temperature and high pressure gasses and expands the hot gasses into the turbine section. The turbine section consists of stationary and rotating blades [7].

As hot gas combustion expands through the turbine, it spins the rotating blades. The rotating blades drive the compressor section to draw more compressed air into the combustion section, as well as generate the electricity power by spinning the generator [8]. The power generation gas turbines are operated at a constant speed to maintain the frequency at the generator output. Baseload is the most favorable operating mode and it has been recommended by Original Equipment Manufacturer (OEM). Under a peak load condition, there are many start–stop modes which can cause the combustor components to undergo a large relative motion, thereby resulting in wear problems [9].

A superalloy is well known as a high-performance alloy, usually used at high temperatures due to its excellent performances. These alloys are excellent in thermal resistance and due to this capability, they are normally used in power generation gas turbines. Nimonic, Hastelloy, and Inconel are some of the examples of the materials that are being used in recent models of power generation gas turbines [10–12].

Kurz et al. [13] revealed that wear is the main degradation of gas turbines, particularly the wear effect on contacting surfaces among components in the system. This wear problem induces significant physical changes onto the surface of the components. Wear is categorized as a high-cost problem because it inflicts permanent damage that disables the refurbishment of the affected components [14].

One of the best techniques to protect the affected surfaces is using a wear-resistant coating. Wear-resistance coatings, also known as hardface coatings, are characterized by their hard properties, among others [15]. The hardness of hardface coatings and their excellent abrasive resistance are important when selecting an appropriate coating for wear protection. Moreover, their thermal stability and high melting temperature are some of the important considerations for applications at elevated temperatures [16]. A pair of hard coatings between two contacting surfaces is one of the best options for wear protection. For this reason, Vencl et al. [17] found excellent wear-resistance results when using hardface-coated surfaces in a gas turbine. Chromium-based hardface coatings are preferable because of their excellent wear-protection ability, particularly at high temperatures [18]. Stellite 6 (a combination

of chromium and cobalt elements) also has excellent wear protection properties, such as good abrasion, corrosion, and erosion resistances, whereas chromium carbide (CrC) has excellent wear and oxidation resistance at elevated temperatures [19,20]. CrC and Stellite 6 are the main hardface coatings used for superalloys.

According to Bernstein [21], the contacting surfaces between two components in a gas turbine, such as a combustion liner and a transition piece or a turbine blade and a ring, continually rub against each other owing to vibrations during operation. These surfaces undergo a large relative motion during start–stop operations. Under this condition of operation, wear severity increases. Considering the high contact pressure and temperature, the components degrade over time, thereby limiting the life of the components. The components undergo relative motion associated with the fretting wear mode [22]. The other type of severe surface degradation is erosion wear. Solid-particle erosion is an important surface damage for gas-turbine components, especially turbine blades [23]. Figure 2 shows the wear problems in the hot-gas-path components of a gas turbine [24].

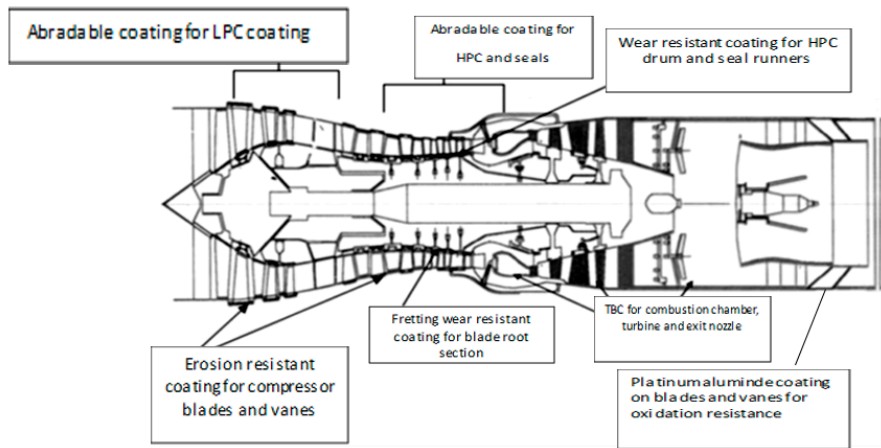

**Figure 2.** Wear problems and hardface coatings for hot gas path components in a gas turbine (Rajendran et al., 2012).

Wear-resistance coatings can be improved by adding a sufficient amount of other hard coating materials. The most important properties of these hardface coatings are excellent wear resistance, thermal stability at elevated temperatures, and resistance to thermal degradations at high temperatures, such as creep, thermal fatigue, and oxidation [25]. Some example metallic elements in a hardface coating that can be incorporated with chromium-based coatings are titanium, tungsten, and zirconium. Surface modifications and treatments of hardface coatings can increase the mechanical properties and thus reduce wear [26].

The main objective of this paper is to summarize wear issues in gas-turbine hot-section components. Wear-resistance coatings are used for combustor–component contacting surfaces, whereas erosion wear-resistance coatings are used for turbine blades and vanes. This review paper presents an overview of a gas-turbine superalloy, characterization of wear, selection of hardface coatings, and current developments in hardface composite coatings for wear protection.

## 2. Gas Turbine Superalloy

Bohidar [3] found that the hot-section components of gas turbines are subjected to extremely high temperatures of 1000 to 1300 °C and experience simultaneous thermal fatigue, creep, and high-temperature wear. Hot-gas-path components are fabricated from high-grade superalloys, such as nickel- and cobalt-based ones. Nickel-based superalloys are well-known metallic alloys with high-temperature applications [27]. The nickel-based superalloys of a gas turbine have some advantages when operated at elevated temperatures. Good strengthening; corrosion, wear, and oxidation resistance; and grain-boundary strengthening are some of the factors that enable these superalloys to maintain

their excellent properties at high temperatures for an extended period [28]. Combustor parts are made from various superalloys, such as Nimonic 263 and Hastelloy X [4]. The less critical components such as fuel nozzles, which are subjected to low or moderate temperatures, are made from the high-strength stainless steel SS 304. High-grade stainless steels are generally fabricated for turbine discs, wheels, and other non-critical components [29].

Hastelloy X is widely used as combustion-section components in gas turbines. This superalloy is one of the best selections for oxidation environments owing to its capability to withstand excessive operation at high temperatures and repeated thermal cycles [30]. With increased operating temperatures up to 1300 °C, the superalloy Hastelloy X is slowly being replaced by Nimonic 263, which is a favorable choice of material for combustor components in gas turbines. This nickel-based superalloy is typically used in high-temperature and high-stress environments that result in combined degradations, such as high-temperature wear, creep, and thermal fatigue [31].

During the operation of peaking load with continuous starting and stopping operations, combustor parts are exposed to sudden changes in temperature, i.e., from ambient to high temperatures of 1300 °C, as well as from 1300 °C to ambient temperature, during a shut-down [32]. Hastelloy X has a melting-temperature range from 1260 to 1355 °C, whereas Nimonic 263 has a melting temperature range from 1300 to 1400 °C. Currently used turbine blades made from strengthened solution and participation-hardened directional solidified and equiaxed GTD 111 [33]. Figure 3 shows a GTD 111 blade used in a gas turbine.

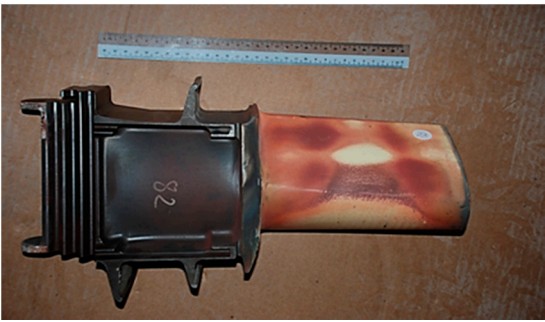

**Figure 3.** Gas turbine blade GTD 111.

Turbine blades, particularly first-stage ones, must withstand the most aggressive environments, thereby limiting the life of the components. Creep life, low cycle fatigue, wear-protection coatings, and oxidation resistance are examples of mechanical properties [34]. For some gas-turbine applications that require a > 700 °C operating temperature, high-strength, and creep-resistant ferritic steels are preferred given their lower cost [35]. Hot-gas-path components are usually fabricated from many different alloy elements and have excellent mechanical properties at high temperatures [36].

Other materials such as FSX 414 and Inconel 738 are primarily used for turbine vanes. These materials have excellent thermal stability, weldability, oxidation resistance, and wear resistance [37]. For example, Inconel 738 is used for third-stage blades because of its lower strength than GTD 111, which is primarily used for first- and second-stage turbine blades. These materials have a combination of a few elements used for an aggressive environment. For example, a combination of chromium and carbon forms carbides with excellent strength in various environments [38]. These carbides also have excellent protection against wear, oxidation, and corrosion. Other examples of materials with a similar strength to these materials are MarM 247 and RENE 80, which are also used for turbine vanes [39]

## 3. Relation of Wear to the Hot-Gas-Path Components of Gas Turbines

Wear significantly affects the hot-gas-path components of gas turbines. The contact surfaces of the main components, such as a combustion liner and a transition piece, as well as turbine blade roots in

contact with their rings, continually rub against each other owing to combustion pulsations during the start and stop operations of gas turbines. These contacting surfaces can undergo large and severe relative motion [40,41].

Tzimasa et al. [42] revealed that under continuous cyclic-duty applications, wear is one of the significant problems experienced by gas-turbine hot-gas-path components. During a hot-gas-path inspection, component disassembly is performed to replace the worn-out components. A proper inspection concentrates on the main components. Wear damage including inspection wear, erosion wear, and cracks due to fretting fatigue occurs on a gas-turbine blade owing to fretting effects during hot-gas-path inspection [43,44]. Figure 4 shows the examples of wear damage on a gas-turbine blade [45,46].

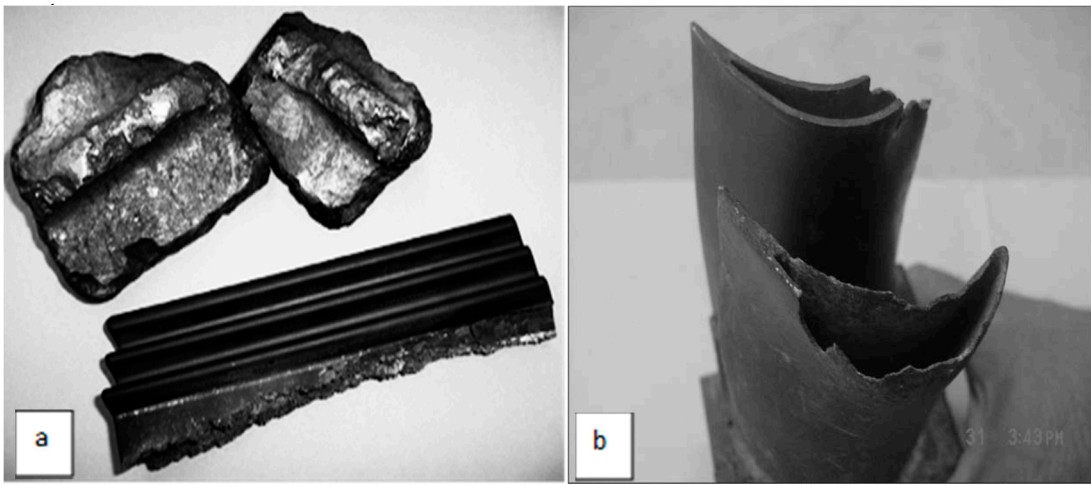

**Figure 4.** Wear damage of gas turbine components: (**a**) fretting wear damage for turbine blade GTD 111 (Barella et al., 2011); (**b**) erosive wear on gas turbine blades (Khajavi and Shariat, 2004).

Kim et al. [47] explained the details of a gas-turbine hot-gas-path inspection. The components of the combustion system are very costly, and significant efforts are exerted to repair the components before scrapping them. Combustor parts and turbine parts are fabricated by structures. During repair, the worn surfaces or other damaged areas can be cut and new pieces can be welded into place. The purpose of repair is to return the component to its original condition, "like new," or "almost new" [48]. The standard process used to repair gas-turbine components are heat treatment, welding, brazing, and recoating [49].

Sahraoui et al. [50] revealed that friction and wear are the most severe degradations undergone by several main components of gas turbines. A similar study was carried out by Schlobohm et al. [51], who found that 70% of the total of hot-gas-path components experience wear problems. Detailed wear characterization is important to investigate the wear mechanisms and to monitor damages during inspection intervals. Chan et al. [52] revealed that fretting wear is one of the main failure modes of hot-gas-path components. These components are subjected to a fretting phenomenon owing to the high vibrations during gas-turbine operations. Damage by fretting-wear mode causes a failure problem, which affects the safety, economy, and lifetime service of hot-gas-path components. Barella et al. [53] showed that a combination of operating parameters influence wear initiation in gas turbines; these parameters include the load of component contact surfaces, movement of surfaces caused by vibration, temperature, wear debris, and materials properties. The appearance of worn out surfaces on hot-gas-path components, regardless of severity, is usually visible [54]. Surface roughness, debris size, friction, and wear rates are amongst the physical considerations to characterize wear severity. For example, severe wear shows a very rough surface in line with the wear track, large delaminated surface, and large debris size, and the friction and wear rates are usually very high [55].

## 4. Relation of Wear to the Hot-Gas-Path Components of Gas Turbines

### 4.1. Wear-Protection Coatings

The use of hardface coatings can significantly increase the wear lives of gas-turbine hot-gas-path components for operating conditions under which the wear protection of these components exceeds the acceptance criteria and limits [56]. The quality of a hardface coating depends on the surface properties of the hardface materials. To correlate the interaction between a hardface coating and its substrate, the compatibility amongst materials, surface roughness, and hardness is important [57]. High hardness and low surface roughness of a hardface coating require wear protection for long operation in gas turbines [58].

Several authors have discussed the use of these protection methods to reduce fretting fatigue wear. CrC is one of the most common choices to reduce fretting because of its capability to delay wear and reduce the formation of wear debris [59]. An anti-fretting coating, Ni-CrC, has been discussed by [60,61]. The addition of nickel binder into the original CrC improves its performance to mitigate wear. The turbine blade root that undergoes shot peening is coated with a copper–nickel coating to produce fretting-resistant surfaces [62]. Figure 5 shows the microstructure of CrC for hard-coating purposes [63].

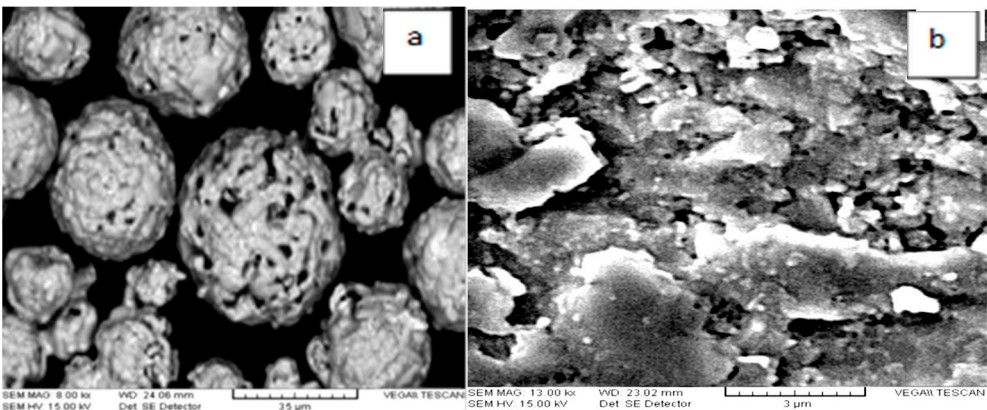

**Figure 5.** Chromium Carbide coating: (**a**) powder form; (**b**) cross sectional image for chromium carbide coating (Yaghtin et al., 2015).

Fang et al. [64] revealed that tungsten carbide (WC) and titanium carbide (TiC), which are usually used in gas-turbine parts, can be subjected to air-plasma spray (APS) and high-velocity oxygen fuel technique (HVOF). The use of these coatings together with melted binders such as nickel and copper is for quick melting through thermal-spraying techniques. WC and TiC have very high hardness (up to 700 HV) as well as the cross sectional section of a titanium-carbide coated surface [65]. The main constraint of using WC is its limited operation in moderate temperatures of up to 600 °C. CrC and TiC coatings have excellent performance up to 1100 °C. WC coatings are widely used at mid-range temperatures. This coating has excellent capability because of its high hardness and strong adhesion capability, rendering it suitable for wear resistance [66]. Figure 6 shows an example of thermal-sprayed TiC for hardface coating [67].

Abradable coatings such as nickel aluminum and aluminum silicon are suitable for gas-turbine components, which experience rubbing modes. The root surface of the turbine blades experiences rubbing with the contact to the casing. The rotating blades experience severe rubbing wear against stationary blades [68]. The significant efforts to minimize the clearance between casing and blade roots using these abradable coatings have resulted in turbine efficiency [69]. Abradable coating plays a very significant role in protecting the blade root, thereby minimizing the clearance between two contacting surfaces, as well as reducing friction between the surfaces. During operation, an abradable coating

remains smooth; it does not produce a significant amount of wear debris nor initiates groove surfaces under all possible rubbing modes [70].

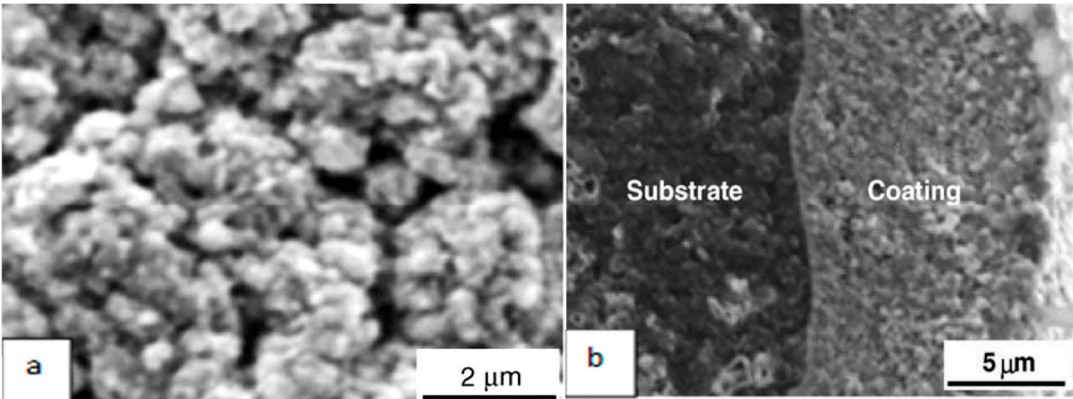

**Figure 6.** Titanium carbide coating: (**a**) powder; (**b**) cross sectional image (Yin et al., 2005).

### 4.2. Erosion-Protection Coatings

Erosion is one of the main degradations usually occurring on turbine-blade surfaces. Material loss from a solid surface because of severe particle impact is called erosion wear. Mechanical action caused by fluid particles is also defined as erosion. The effect of erosion wear is significant when it comes to gas-turbine efficiency. The turbine blades of hot-gas-path components are subjected to efficiency loss because of the impact of solid-particle erosion on blade surfaces [71].

Erosion-resistant coatings are introduced to reduce the damage caused by particle impact. The continuous process of erosion on the surfaces of hot-gas-path components causes severe material loss that can lead to failure. Examples of erosion-resistant coatings are titanium-based ones such as TiC and titanium nitride [72–74]. With an average hardness of 700 HV and thickness of 50 μm, these coatings can reduce the particle impact, which can lead to cracks or premature failures. These coatings provide a protective layer to reduce the continuous propagation of particle impact.

WC–Co–Cr hardface coating also has excellent erosion resistance as a hard coating. This coating has shown a consistently low erosion rate because of its high hardness and toughness. WC is applied to a gas-turbine blade to reduce the friction between blade surfaces and wear particles. Moreover, this coating is used to increase the durability of the blade surface and prolong the service life of components by enhancing surface properties [75,76].

Fe–Cr coating is built over a bond coat of a turbine blade to prevent the leading and trailing edges from significant, premature material loss because of erosion. The overall thickness layer of Fe–Cr coating is approximately 3–25 μm. Owing to the severe impact of hard erosion particles, Fe–Cr suffers significant material loss, which leads to crack formation. The remaining thickness of the eroded surface is between 1–10 μm [77].

In general, research on erosion-resistant coatings is significant. Severe mechanical attack from solid particles reduces the original dimension of the hot-gas-path components of gas turbines [78]. Several cases of fluid erosion or water-droplet erosion have been reported; however, solid-particle erosion plays a significant role in the occurrence of premature failures of gas-turbine hot-gas-path components. Power plants located in dusty and sandy environments are prone to erosion problems. Premature failures are important because of erosion, which causes cracks [79,80].

### 4.3. Current Developments in Combination Wear-Resistant Coatings

The APS technique is the most widely used thermal-spray technique for hardface coatings in gas turbines. Fretting fatigue decreases the wear and fatigue strengths of gas-turbine materials, particularly for cyclic load operations. In fretting fatigue, microcracks and cracks can be initiated

at the fretting zones, and these cracks propagate into the substrates [81]. To mitigate this problem, advanced hardface coatings for hot-gas-path components should be designed to have a dual function, namely, high hardness and low porosity to reduce wear, and the ability to reduce wear debris trapped under the fretting bridge [82]. Current hardface coatings are usually designed for baseload conditions. However, hardface coatings are subjected to wear damage under severe peaking conditions, thereby limiting the wear lives of the components [83]. Advances are needed for these hardface coatings to be sustained under peaking-load condition, including their thermal-spray parameters. The selection of appropriate hardface coatings is important to ensure that the coating is excellent for wear protection and can adhere very well onto a substrate [84].

Combining hard metallic coatings is one of the best methods of developing excellent hardface coatings for wear resistance. With prolonged reaction time and increased firing temperatures of the gas turbine up to 1300 °C, suitable hardface coatings with the abilities to withstand operation at this temperature and be operated under cyclic load are important to find [85]. Currently, minimal research has focused on mitigating fretting fatigue wear in gas turbines. As an advanced thermal-spray technique, HVOF has been introduced to produce low porosity and high-hardness coated surfaces [86].

Titanium nitride–titanium has excellent erosion resistance for blade surfaces. However, this material's working temperatures is limited only to <800 °C [87]. Xu et al. [88] developed an aluminum-based composite coating ($Al_2O_3$), i.e., Stainless Steel 304, for gas-turbine combustor materials. This composite coating can be used only for high-grade stainless steels. The average hardness of the coating is 600 HV, which renders it suitable for wear resistance. $Al_2O_3$ coating also has excellent wear and erosion resistance and can be used at low temperatures. The fuel nozzle of a gas turbine can be made from SS 304, and the operating temperature of this component is below 500 °C.

Stellite 6 coating is one of the main hardface coatings used in gas turbines. Bartkowski et al. [89] prepared Stellite 6 and WC (S6/WC) composites through a laser-cladding technique to improve the wear, hardness, and corrosion resistance of metals. The average hardness is 350 to 650 HV. The composite coating shows strong bonding between the main carbide particles of $M_7C_3$, $M_6C$, and $M_{23}C_6$. Similarly, Li et al. [90] prepared solid-state WC reinforced on Stellite 6 coating to improve the tribological property of a new coating. A higher WC content in Stellite 6 increases the wear resistance and thus reduces the tendency of surface-cracking initiation. WC addition shows excellent ability to increase the wear-resistance performance; however, WC is well-known for its limited performance at temperatures below 700 °C.

Wang and Zui [91] fabricated HVOF-sprayed CrC by adding metal cermet. They further investigated the erosion behavior of several candidates of CrC–NiCr particles. The composite powder is found to have a lower chromium content and a higher nickel content than CrC coating. The composite also excellently performs for erosion-wear resistance as the coatings have high microhardness, fine microstructure after HVOF spraying, low oxidation, and low porosity.

Szymanski et al. [92] compared several candidate hardface coatings used for power generation. They are WC–Co, WC–Co–Cr, WC–CrC–Ni, and Cr2Cr3–NiCr. Tribological-property studies indicate that WC–Co–Cr has the highest hardness amongst all candidates, i.e., between 1250 and 1400 HV. Thus, this hardface coating has excellent cavitation resistance and corrosion resistance. The powder is produced through various techniques. Figure 7 shows the powders produced for the comparative study.

The rule of thumb for a coating combination is to produce coatings with properties similar to those of the substrate. NiCrCoAlY and CoNiCrAlY coatings have similar thermal expansion coefficients to the nickel-based superalloy for the hot-gas-path components of gas turbines. These coatings have excellent oxidation and corrosion resistance at elevated temperatures. The recommended thermal-spray processes are APS and HVOF as these methods induce high bond strength after coating [93]. Figure 8 shows the powder feedstock for NiCrCoAlY and CoNiCrAlY.

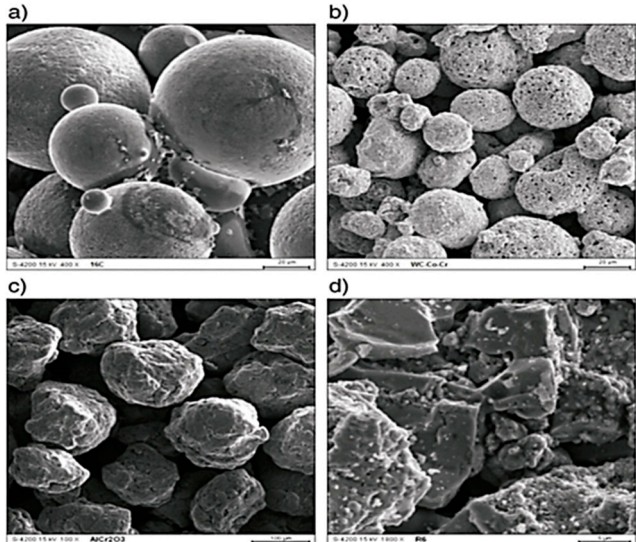

**Figure 7.** Titanium Carbide coating on austenitic steel produced by the way of: (**a**) spraying, (**b**) agglomerating and sintering, (**c**) mechanical alloying, (**d**) high temperature synthesis (Szymanski et al., 2014).

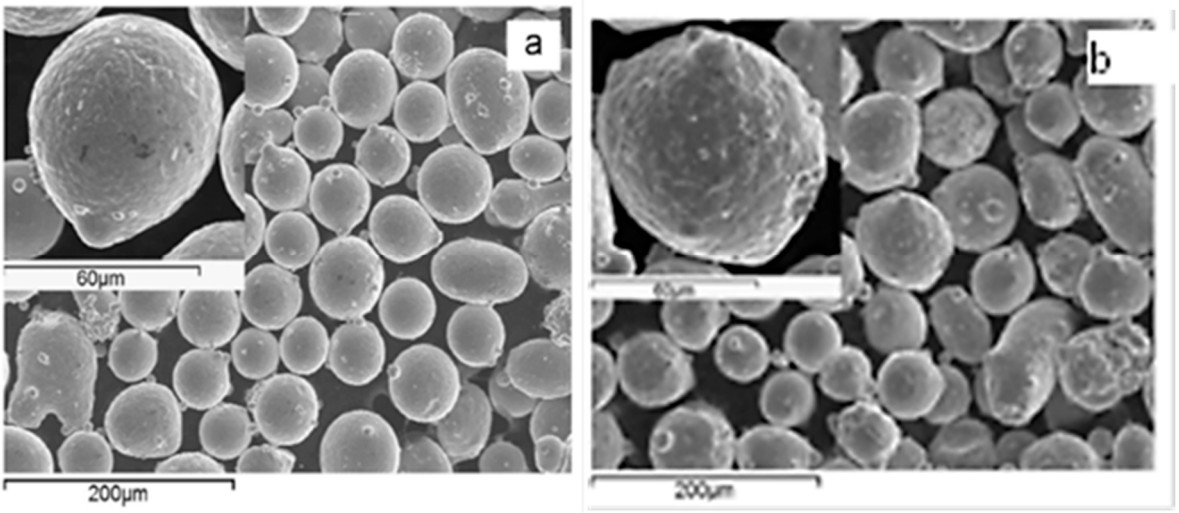

**Figure 8.** Microstructure of powder feedstock: (**a**) NiCrCoAlY and (**b**) CoNiCrAlY (Pereira et al., 2015).

## 5. Conclusions

This review paper presents a detailed discussion of wear problems in hot-gas-path components and various hardface coatings. Wear is concluded as one of the main problems in gas-turbine hot-gas-path components. Erosion-resistant coatings can improve the erosion life of hot-gas-path components, particularly turbine blades. Fretting-wear-resistant coatings protect the contact area between two components from fretting fatigue, wear, and crack initiation.

Hardface coating is the most valuable coating process to improve the life of the worn-out components in gas turbines. Recently, this process has been widely chosen for wear reduction and wear replacement of the components. This application of hardface coating has the advantages of reducing wear as well as increasing the life of the components. With respect to coating selection, Chromium Carbide and Stellite 6 are the most popular for gas turbine components while other coatings are being used for some low-temperature applications. The cost effectiveness of the hardface coating and hardface materials depend on specific applications based on materials and elemental composition for the particular area to be protected.

**Author Contributions:** Conceptualization, A.A.P., M.J.G. and A.R.; methodology, A.A.P.; validation, A.A.P., M.J.G. and A.R.; formal analysis, A.A.P., M.J.G. and A.R.; investigation, A.A.P., M.J.G. and A.R.; resources, A.A.P., M.J.G. and A.R.; writing—original draft preparation, A.A.P.; writing—review and editing, M.J.G., W.F.H.W.Z. and A.R.; visualization, A.A.P.; supervision, M.J.G., W.F.H.W.Z. All authors have read and agreed to the published version of the manuscript.

**Funding:** This research received no external funding.

**Acknowledgments:** This study was supported by the TNB Research Sdn Bhd (Malaysia) and The National University of Malaysia (UKM).

**Conflicts of Interest:** The authors declare no conflict of interest.

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
