# Peer review of "Wear Characteristics of Superalloy and Hardface Coatings in Gas Turbine Applications–A Review"

_metals, doi:10.3390/met10091171_

Round 1
Reviewer 1 Report
The topic of the manuscript “ Wear Characteristics of Superalloy Materials and Hardface Coatings in Gas Turbine Applications – A Review” seems to be interesting and important. In my opinion, this study is a valuable work, the paper is well written, the research is well designed. However, I have some doubts about the paper which I listed below; therefore I suggest this paper for “minor revision”:
- Since the scope of this journal is not the gas turbines, the authors could introduce a bit more deeply this interesting application for the readers! As well as they should define the superalloys. In the case of wear-resistance, and mainly in the case of erosion, it would also be crucial to define what do you mean exactly in the given case. E.g. I could tell you at least 10 different types of erosion.
- The quality of the figures is generally low. I know that it is a hard issue in the case of review papers since the authors need to use what they found, but they could improve at least some parts. E.g. they could redraw the captions in the case of the SEM figures. Most of them are barely visible.
- The Conclusions section is too short and a bit meaningless. Please provide more detailed conclusions according to your deep review.
Reviewer 2 Report
Conclusions too general. The assumption of the publication is that "Wear is concluded as one of the main problems in gas-turbine hot gas-path components". Authors should show the relationship between materials or coatings and the intensity of wear processes.
Reviewer 3 Report
The paper is a review of the wear problems of superalloys used in hot-gas-path components. Various coatings to deal with this problem are also reported. the paper is fairly comprehensive and gives a good insight regarding superalloys for turbines and their wear problems. However some improvements are recommented.
1)The paper needs some editing to correct minor typo or syntactic mistakes. For example the authors refer to all figures as Figure A1, A2 etc.
2) A more detailed visual display of the wear problems encounted in such systems would help the readability of the paper. Presentation of more figures (not anly figure 4) depicted such wear problems would give an added-value to the review paper.
3) It would be usefull to add some cross-sectional images illustrating the microstructure of the hard coating used. Detailed analysis of the phases developed with respect to the production process of the coating, along with advantages and drawbacks for each case would be useful information.
Round 2
Reviewer 3 Report
The authors have manage to sucessfully cope with the remarks proposed. The paper can be accepted for publication.
Author Response
Well noted. Thank you